# Intertwined risk factors of mental health and cardiovascular diseases: A cross-sectional survey in Godawari Municipality of Far-western Nepal

**Ramesh Ojha, Chiranjivi Adhikari** [ID]*, **Hari Prasad Kaphle** [ID], **Dikshya Adhikari**

School of Health and Allied Sciences, Pokhara University, Kaski, Nepal

* chiran.adhikari@pu.edu.np

## Abstract

### Introduction

Cardiovascular diseases (CVDs) are the leading global cause of death, whereas mental disorders account for one-third of all global disabilities. Despite clear available evidence of a significant relationship between the CVDs and mental disorders, and the CVDs being regressional to certain risk factors, they are less explored in the Nepalese context, especially for mental disorders. Hence, we aimed to determine the prevalence of mental health status and its associated factors among people with cardiovascular risk status.

### Methods

A community-based study was carried out during Sep-Nov 2024 among 390 people aged 30–69 years with cardiovascular risk status in Godawari Municipality, Kailali. Data were collected with face-to-face interviews using structured questionnaires consisting of four sections: a) socio-demographic characteristics; b) CVD risk factors; c) mental health status; and d) anthropometric measurements, using the KoboToolbox, a mobile and computer-based application. Further, we imported data into SPSS software for statistical analysis. We presented categorical variables as frequency and percentage and continuous variables as median and quartiles. We applied univariate and multivariate logistic regression to determine factors associated with depression symptoms, anxiety, and stress. The result of logistic regression was presented as crude odds ratio, adjusted odds ratio (AOR), beta coefficients (β), their 95% confidence interval, and p-values.

**Data availability statement:** All relevant data are within the manuscript and its Supporting information files.

**Funding:** The author (s) received no specific funding for this work.

**Competing interests:** The authors have declared that no competing interests exist.

## Results

Of 390 participants, females were two-thirds (67%), and the median age was 48 (36~60). The prevalences of depression, anxiety, and stress symptoms were found to be 47.2%, 62.3% and 55.1%, respectively. From multivariate logistic regression analysis, depression symptoms were positively associated with females (β = 1.002, p < .001) and the presence of one CVD risk factor (β = 1.082, p = .016), two risk factors (β = 1.362, p = .006) and three or more risk factors (β = 1.720, p = .017). Anxiety symptoms were associated with exposure to secondhand smoking (β = 0.725, p = .024) and the presence of one risk factor (β = 1.548, p < .001), two risk factors (β = 1.734, p < .001) and three or more risk factors (β = 1.852, p = .022). Dalit, Janajati and Madhesi (β = 0.735, p = .026) and the presence of one risk factor (β = 1.811, p = .001), two risk factors (β = 2.054, p = .016) and three or more risk factors (β = 2.138, p < .001) were associated with stress symptoms. All three mental disorders were found to worsen as the number of CVD risk factors increased.

## Conclusion

The study revealed that nearly fifty percent of the prevalence of each of depression, anxiety, and stress symptoms among people with cardiovascular risk status. Mental health screening is suggested among the people with CVD risk factors, additionally considering female, secondhand smoker, and disadvantaged ethnic populations for depression, anxiety, and stress, respectively. Further, a validation study is recommended for accuracy and yield.

## Introduction

Cardiovascular diseases (CVDs) are the global public health problem, causing 17.9 million deaths in 2019, and among them, more than three-quarters of deaths occur in low- and middle-income countries [1,2]. CVD-related deaths have increased by 12.5% over the last decade and are recognized as the most significant concerns in public health [3]. Nepal is also facing an increasing burden of CVDs, with a 3.8 case fatality rate and 1,104,474 disability-adjusted life years in 2019 [4]. Sudurpashchmin Province had a significantly higher percent (9.8%) of adults aged 40–69 with 30% or more CVD risk than almost all other provinces [5]. Smoking and tobacco use (33.7%), alcohol consumption (20.8%), unhealthy diet (>90%), and hypertension (21%) were established risk factors for CVDs [5]. These factors often lead to mental health morbidity and mortality. In 2019, 970 million people worldwide suffered from a mental disorder, the most frequent of which were anxiety and depression [6]. According to NDHS 2022, the prevalence of symptoms of anxiety and/or depression in Far-western Nepal is 24.6% (urban 28.3% and rural 18.4%) [7].

There is a bidirectional relationship between mental health and CVDs [8]. Mental health issues are more common in people with cardiovascular disease and its risk factors than in healthy people [9] and severe mental illness has been linked to a

two-fold increase in deaths from CVDs [10]. A prospective cohort study from the UK demonstrated that the highest mental health scores were associated with a 1.56-fold increase in CVD risk [11]. A systematic review found that mental health conditions like depression and anxiety may cause fluctuations in blood pressure, potentially leading to cardiovascular complications [12]. Research on mental health is often given less emphasis, despite the fact that depression and anxiety are predicted to cost the global economy $1 trillion a year [13].

Evidence from recent studies showed that mental health is increasingly recognized as a major factor in cardiovascular disease risk, and integrating mental health assessments into cardiovascular care is essential for reducing the disease's impact. A Mendelian randomization carried out with a very large sample strongly evidenced that smoking is a causal factor for depression and schizophrenia [14]. Similarly, multi-cross-sectional findings from a cohort study of patients and their siblings also evidenced that smoking is associated with both positive and negative psychosis, and depression [15]. There are also strong findings from evidence syntheses. A metaanalysis showed 70% more odds of having severe mental illness (SMI), and it increases to 87% in the case of schizophreniform illness, among the diabetics, and also, metabolic syndrome (MetS) is found in higher proportion among the people with SMI [16]. Inversely, another review showed that those with SMIs, 70% of their diabetic statuses remain unscreened [17]. The significant relationship of these two giant co-morbidities, and their risk factors, is complex, intertwined, multi-faceted, and magnitudinous, and further implicates healthcare delivery and outcomes, as in metaanalysis carried out by Ayerbe et al. revealed that odds of quitting smoking is 36% less among the depressed patients [18]. However, these relational factors are less explored in Nepalese context, and even less in Far-western Province. Hence, the study aimed to assess the prevalence and risk factors associated with mental health status (depression, anxiety, and stress symptoms) among people with cardiovascular risk status.

## Methods

### Study design, population, and setting

We conducted a cross-sectional study from September 1 through Nov 15, 2024, among participants aged 30–69 years with cardiovascular risk status in Godawari Municipality of Far-western Nepal.

### Sample size and sampling technique

The sample size was calculated using Cochran's formula with finite population correction with a 5% margin of error and 95% confidence level, and taking the prevalence of mental health status of 50%. The calculated sample size was 384, but we ultimately enrolled 390 participants applying the Probability Proportional to Size (PPS) sampling method. Out of 12 wards, three were selected randomly using lottery method. From the selected wards, the required number of individuals were calculated based on PPS and then, households were selected conveniently. If more than one eligible participant was present in the household, we selected one randomly. The inclusion criteria for the study were people aged 30–69 years and who were permanent residents and living there for last 6 months in Godawari Municipality, Kailali. The exclusion criteria included: a) people who were ill and unable to communicate during the data collection period, b) people with known and diagnosed cardiac disease, or using stent, c) pregnant women, and d) mentally retarded and bed-ridden patients.

### Data collection

We collected data with face-to-face interview using a structured questionnaire consisting of four sections: a) socio-demographic characteristics; b) CVDs risk factors; c) mental health status; and d) anthropometric measurements. We collected data using electronic forms in the KoboToolbox, a free and open-source online data entry mobile application developed by Harvard Humanitarian initiatives [19]. The participants were oriented about the purpose of the study before data collection. It took about 30–35 min for each participant to complete the interview.

## Variables and measurement

Mental health status: It was defined as the current state of depression, anxiety, and stress symptoms among the study participants. Nepali version of the validated Depression Anxiety and Stress Scale-21 (DASS-21) [20] was used to measure mental health status. The internal consistency (Cronbach's alpha) of the DASS-21 Nepali version was 0.93 for DASS-depression; 0.79 for DASS-anxiety; and 0.91 for DASS-stress [21]. DASS-21 is a widely used and validated tool for measuring mental health outcomes in many countries, including Nepal [22,23]. There are 21 items on the DASS-21 scale (7 items each for depression, anxiety, and stress symptoms). Each participant is asked to score every item on a scale from 0 to 3, where 0 indicates "did not apply to me at all" and 3 indicates "apply to me at all". The sum of each scale was multiplied by two to determine the overall scores for depression, anxiety, and stress symptoms. Depression, anxiety, and stress symptoms were further classified as binary (present/absent) if at least mild conditions were present [22].

The independent variables assessed in this study included age (in years), gender (male/female), ethnicity (Dalit/Janajati/Brahmin/Chhetri), education status (illiterate/informal education/basic level/secondary level/bachelor's or above), marital status (unmarried/married/wife), main occupation (unemployed/private employee/laborer/housekeeper/farmer/government job/business), and monthly family income (in NPR), smoking status (never/former smoker/current smoker), smokeless tobacco (never/former chewer/current chewer), frequency of consumption of smokeless tobacco (daily/sometimes), secondhand smoking (yes/no), fruits and vegetable intake (adequate/inadequate), body mass index status (underweight/normal/overweight/obese), blood pressure status (normal/hypertensive), body fat status (normal/excess), basal metabolic rate (lower/ideal), and hand grip strength (lower/ideal).

CVDs risk factors were defined as the presence of one or more risk factors in each participant, and these were assessed by counting the risk factors from current smoking and tobacco use, inadequate intake of fruits and vegetables, overweight/obesity, hypertension, and excess body fat. Thus, their presence was assigned a number between zero to five. Smoking and tobacco use [24], fruits and vegetables consumption [8], body mass index [25], blood pressure [26], body fat [27], basal metabolic rate [28], and hand-grip strength [29] were defined using the protocol given by previous research. Fruit and vegetable consumption was further divided into low and high, taking the median as a cut-off point.

## Statistical analysis

The KoboToolbox platform was used for data collection and entry [19]. The data was cleaned and then exported to Statistical Package for Social Sciences (SPSS) version 25 for further analysis. Continuous variables were presented as median, and quartiles after checking for normal distribution whereas categorical variables were presented as frequencies and percentages. The normality of continuous variables was displayed visually (histogram) and numerically (Kolmogorov-Smirnov tests and Shapiro-Wilk test). Chi-square test (p-value <0.05 at 95% level of confidence) was used for testing for association between the dependent and independent variables (nominal and ordinal), and Mann Whitney U test statistic (p-value <0.05 at 95% level of confidence) was used for testing for association between the dependent and independent variables (continuous). We applied binary logistic regression to model the associations between CVD risk factors and mental health outcomes, accounting for potential confounding factors and providing adjusted odds ratios to quantify these relationships in binary logistic regression, we incorporated all variables with a chi-square test p-value below 0.25. Crude and adjusted odds ratio, beta coefficients (β) with 95% confidence intervals were calculated, and p-values < 0.05 were used to determine whether a variable was statistically significant. The strength of association of independent and dependent variables was calculated by comparing associations obtained sequentially from multiple models of zero, one, two, and more than two risk factors of CVDs with mental health statuses. Before conducting the logistic regression, the Durbin-Watson test and variance inflation factor (VIF) with tolerance statistics were carried out to check the independence of errors and multicollinearity of the independent variables and revealed no multicollinearity. Parameters such as Negelkarke R squares, Cox and Snell R squares, the log likelihoods, and the Hosmer and Lemeshow test were carried out to determine the model's goodness of fit and the variance of independent variables that described the mental health status.

### Ethical consideration

We obtained ethical approval from the Institutional review committee of Pokhara University (Reference Number: 96/2081/82-IRC, approval date: August 30, 2024). Formal permission was obtained from Godawari Municipality before conducting the study. We obtained written informed consent from each participant before enrolling them in the study. We ensured that participation in the study was voluntary and maintained confidentiality throughout the study.

## Results

### Socio-demographic, behavioral, and bio-physical characteristics of the study participants

Of 390 participants, two-third (66.9%) of them were females, with Brahmin/Chhetri as a major ethnic group (83.8%). Surprisingly, 11.8% were widowers or widows. One in every five participants was illiterate. Among total participants, about one-third (32.1%) of them were overweight and obese. Nearly one third (30.3%) of respondents were hypertensive, whereas less than half (40.5%) of the participants had excess body fat (Table 1)

### Prevalence of number of CVDs risk factors among study participants

Furthermore, we assessed the prevalence of risk factors for CVDs risk factors by the number of factors across sexes and different age groups. The median risk factor ($Q_1 \sim Q_3$) was 2 (1~3). Majority (88.2%) of them had one or more CVDs risk factors (Fig 1).

### Prevalence of mental health status (DASS 21 score)

The prevalence rates of depression, anxiety, and stress were found to be 47.2%, 62.3%, and 55.1% among people with cardiovascular risk status (Table 2).

Participants constituting 6.9%, 11.0%, and 7.7% experienced only depression, anxiety, and stress symptoms, respectively, whereas 29.2% experienced all three mental health issues concurrently (Fig 2).

### Factors associated with depression symptoms

Multivariate analysis showed that sex, and the distribution of CVDs risk factors were statistically significantly associated with depression symptoms among people with cardiovascular risk status. Female participants are nearly three times (aOR: 2.725, 95% CI: 1.575 to 4.715) more likely to have depression symptoms as compared to males. In comparison with zero risk factor, individuals with one risk factor (aOR: 2.952, 95% CI: 1.224 to 7.121), two risk factors (aOR: 3.904, 95% CI: 1.480 to 10.299) and three or more than three risk factors (aOR: 5.585, 95% CI: 1.357 to 22.991) were more likely to have depression symptoms (Table 3).

### Factors associated with anxiety symptoms

Participants who were exposed to secondhand smoking were two-folds (aOR: 2.065, 95% CI: 1.100 to 3.877) more likely to have anxiety symptoms compared to those who were unexposed to secondhand smoking. In comparison with zero risk factor, participants with one risk factor (aOR: 4.703, 95% CI: 2.046 to 10.811), two risk factors (aOR: 5.664, 95% CI: 2.190 to 14.646) and with three or more than three risk factors (aOR: 6.374, 95% CI: 1.314 to 30.915) were more likely to have anxiety symptoms (Table 4)

### Factors associated with stress symptoms

From multi-variate logistic regression, Dalit, Janajati, and Madhesi participants were 2 times (aOR: 2.085, 95% CI: 1.091 to 3.985) more likely to have stress symptoms compared to those Brahmin/Chhetri. In comparison with zero risk factor, participants with one risk factor (aOR: 6.119, 95% CI: 2.202 to 17.008), two risk factors (aOR: 7.800, 95% CI: 1.463 to

**Table 1. Socio-demographic, behavioral, and bio-physical characteristics of participants.**

| Variables (n = 390) | Frequency (n) | Percentage (%) |
|---|---|---|
| **Age (years)** | | |
| 30-39 | 116 | 29.7 |
| 40-49 | 87 | 22.3 |
| 50-59 | 72 | 18.5 |
| 60+ | 115 | 29.5 |
| Median ($Q_1 \sim Q_3$); [Min~Max] | 48 (36~60); [30~69] | |
| **Sex** | | |
| Male | 129 | 33.1 |
| Female | 261 | 66.9 |
| **Ethnicity** | | |
| Brahmin/Chhetri | 327 | 83.8 |
| Dalit | 40 | 10.3 |
| Janajati | 23 | 5.9 |
| **Educational level** | | |
| Illiterate | 75 | 19.2 |
| Informal education | 101 | 25.9 |
| Basic level | 114 | 29.2 |
| Secondary level | 87 | 22.3 |
| Bachelor's or above | 13 | 3.3 |
| **Marital status** | | |
| Unmarried | 3 | 0.8 |
| Married | 341 | 87.4 |
| Widower or widow | 48 | 11.8 |
| **Main occupation** | | |
| Housekeeper | 124 | 31.8 |
| Farmer | 97 | 24.9 |
| Unemployed | 91 | 23.3 |
| Labor | 30 | 7.7 |
| Private employee | 21 | 5.4 |
| Business | 14 | 3.6 |
| Government job | 13 | 3.3 |
| **Smoking status** | | |
| Never | 289 | 74.1 |
| Former smoker | 35 | 9.0 |
| Current smoker | 66 | 16.9 |
| **Smokeless tobacco consumption** | | |
| Never | 298 | 76.4 |
| Former chewer | 12 | 3.1 |
| Current chewer | 80 | 20.5 |
| **Exposure to secondhand smoking** | | |
| Yes | 139 | 35.6 |
| No | 251 | 64.4 |
| **Fruits and vegetables consumption** | | |
| < 1 serving per day | 10 | 2.6 |
| 1–2 servings per day | 290 | 74.4 |
| 3–4 serving per day | 48 | 12.3 |

*(Continued)*

**Table 1.** (Continued)

| Variables (n = 390) | Frequency (n) | Percentage (%) |
|---|---|---|
| ≥ 5 serving per day | 42 | 10.8 |
| Median (Q$_1$~Q$_3$); [Min~Max] | 1.3 (1.1~2.1); [0~6] | |
| **BMI status** | | |
| Underweight (BMI < 18.5 kg/m2) | 35 | 9.0 |
| Normal (BMI 18.5–24.9 kg/m2) | 230 | 59.0 |
| Overweight and obese (BMI ≥ 25 kg/m2) | 125 | 32.1 |
| **Blood Pressure Status** | | |
| Normotensive (<140 mm/Hg and/or <90 mm/Hg) | 272 | 69.7 |
| Hypertensive (≥140 mm/Hg and/or ≥90 mm/Hg) | 118 | 30.3 |
| **Body Fat (n = 390)** | | |
| Normal | 232 | 59.5 |
| Excess | 158 | 40.5 |
| **Basal metabolic rate (n = 390)** | | |
| Lower | 278 | 71.3 |
| Ideal | 112 | 28.7 |
| **Hand grip strength (n = 390)** | | |
| Lower | 143 | 36.7 |
| Ideal | 247 | 63.3 |

41.582) and participants with three or more than three risk factors (aOR: 8.484, 95% CI: 2.769 to 25.997) were more likely to have stress (Table 5).

## Mental health status and strength of predictors

Distribution of CVD risk factors were significantly associated with mental health status. As the number of risk factors increases, depression, anxiety, and stress symptoms also increase significantly (Table 6 and Fig 3).

## Discussion

From this study, prevalence of depression, anxiety, and stress were found to be 47.2%, 62.3%, and 55.1%, respectively. Firstly, sex, and CVDs risk factors were revealed as risk factors of depression symptoms; Secondhand smoking, and CVDs risk factors were identified as risk factors of anxiety symptoms; and ethnicity, and CVDs risk factors were found as risk factors of stress symptoms. Previous studies identified smoking habits, blood pressure, and body fat were associated with depression, anxiety, and stress symptoms but these are found to be non-significant in this study. We further discuss the prevalence, risk factors, and non- significant factors.

The prevalence of depression, anxiety, and stress symptoms in this study is nearly consistent with the findings reported by Thagunna and his colleagues among the Nepalese youth population, with 50.6% depression, and 56.2% stress [30]. In contrary, the prevalence of depression and anxiety symptoms in this study was notably higher than the 24.6% prevalence of anxiety and/or depression reported in the NDHS 2022 among adults' populations of Far-western Nepal [7]. However, anxiety prevalence of 62% in our study is unmatched and higher than as reported by Thagunna and his colleagues (46.5%) [30], the generalized anxiety of school adolescents (35%) [31], and depression (38%) and anxiety (47%) among undergraduate university students [32]. This was due to the sample population taken in NDHS survey was age group 15–49 years, and more than half of the women and men interviewed were under age 30, and in school adolescents as well as college youths, but in this study the sample population was aged 30–69 years. The prevalence of stress symptoms was consistent with findings reported by the NCD-STEPS Survey 2019 with 62.3% stress [5]. Similarly,

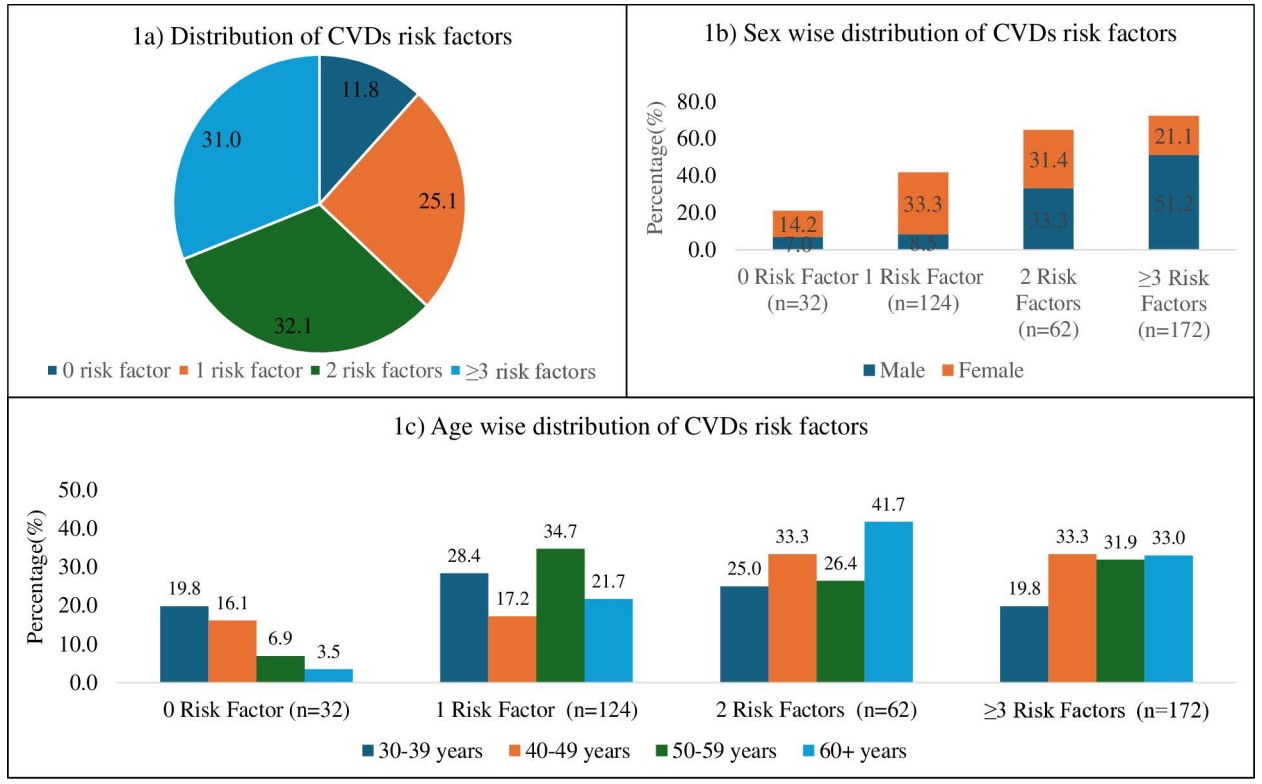

**Fig 1. Prevalence of number of CVDs risk factors among study participants (n = 390).**

the findings of this study are consistent with the findings presented by the study among chronic disease patients in Karnatak, India [9]. A study conducted by Basnet and his colleagues among residents of Nepal reported 35.1% depression, 31% anxiety and 22.5% stress, which are lower than in this study [33]. One convincible reason for this variation may be that CVDs risk factors are significantly associated with mental health status and people with these risk factors are at higher risk of developing poor mental health. The prevalence of depression, anxiety, and stress symptoms was higher than the findings reported by a study conducted among construction workers in Kavre district, Nepal [22], and among traffic police officers in Kathmandu [23]. Our findings indicate a higher prevalence of mental health problems among individuals with cardiovascular disease risk factors in Nepal compared to studies in India [34], Northwest Iran [35], and Ghana [36], suggesting that unique contextual factors may be at play. The possible reasons for the variation in the prevalence of mental health status may be due to differences in the study setting, study period, study population, study variables, socioeconomic status, sample size and sampling techniques, study tool and techniques, data analysis methods, and COVID-19 pandemic.

The findings of this study revealed a group of CVD risk factors was significantly associated with depression, anxiety, and stress symptoms. Findings of previous studies [37,38], showed that people with CVDs risk factors have a greater risk of mental health disorders compared to the general population The possible justification could be that prior studies reported that many CVDs risk factors, such as smoking habits, harmful use of alcohol, hypertension, diabetes, excess body fat, low intake of fruits and vegetables, and overweight and obese were more prevalent. Risk factors of CVDs were significantly associated with mental health status and people with these risk factors are at higher risk of developing poor mental health such as depression, anxiety, and stress. CVD risk factors, such as obesity, excess body fat, and

**Table 2. Prevalence of mental health status (DASS 21 score).**

| Variables (n = 390) | Frequency (n) | Percentage (%) |
|---|---|---|
| **Depression (Score of Depression Subscale)** | | |
| Normal (0–9) | 206 | 52.8 |
| Mild (10–13) | 91 | 23.3 |
| Moderate (14–20) | 59 | 15.1 |
| Severe (21–27) | 32 | 8.2 |
| Extremely severe (28+) | 2 | 0.5 |
| Total with depression symptoms | 184 | 47.2 |
| Median (Q$_1$~Q$_3$); [Min~Max] | 14 (16~22); [6~38] | |
| **Anxiety (Score of Anxiety Subscale)** | | |
| Normal (0–7) | 147 | 37.7 |
| Mild (8–9) | 108 | 27.7 |
| Moderate (10–14) | 70 | 17.9 |
| Severe (15–19) | 60 | 15.4 |
| Extremely severe (20+) | 5 | 1.3 |
| Total with anxiety symptoms | 243 | 62.3 |
| Median (Q$_1$~Q$_3$); [Min~Max] | 9 (7~14); [4~30] | |
| **Stress (Score of Stress Subscale)** | | |
| Normal (0–14) | 175 | 44.9 |
| Mild (15–18) | 67 | 17.2 |
| Moderate (19–25) | 97 | 24.9 |
| Severe (26–33) | 49 | 12.6 |
| Extremely severe (34+) | 2 | 0.5 |
| Total with stress symptoms | 215 | 55.1 |
| Median (Q$_1$~Q$_3$); [Min~Max] | 9 (8~13); [0~30] | |

hypertension, are associated with chronic low-grade inflammation. This inflammation may also impact the brain, possibly changing how brain chemicals work and affecting brain function—both of which can play a role in development of depression, anxiety, and stress. Similarly, behaviour risk factors such as inadequate intake of fruits and vegetables, along with smoking and tobacco use, can negatively affect brain health by lowering essential nutrients and raising harmful toxins. This can impact mood, thinking, and emotional balance, leading to mental health issues. Over time, these behaviors can create a cycle of declining physical and mental well-being.

We critically examined gender disparities, noting that female participants were nearly three times more likely to experience depression symptoms compared to males. This finding is consistent with secondary analysis of NDHS 2022, which revealed a higher prevalence among females (5.4%; 95%CI: 4.8, 6.1) than males (1.7%; 95% CI: 1.4, 2.3) [13]. Studies from China [39], Easten Ethiopia [40] and Ghana [36] also supported this findings and reported that the prevalence of depression was higher in females than males. Although the reason for the higher prevalence of depression symptoms among women is not clearly understood. The possible reasons for this association may be that women are frequently expected to shoulder more caregiving and household responsibilities and suffer disproportionately from gender-based violence, such as sexual and domestic abuse, which raises their risk of depression. The hormonal fluctuations that occur during menstruation, pregnancy, the postpartum period, and menopause can increase the vulnerability of women to mood disorders and consequently to depression symptoms [41,42].

Exposure to secondhand smoking is found to be significantly associated with anxiety symptoms but not with depression or stress symptoms. A cross-sectional study [34] and systematic review [35] have supported this. The possible reason

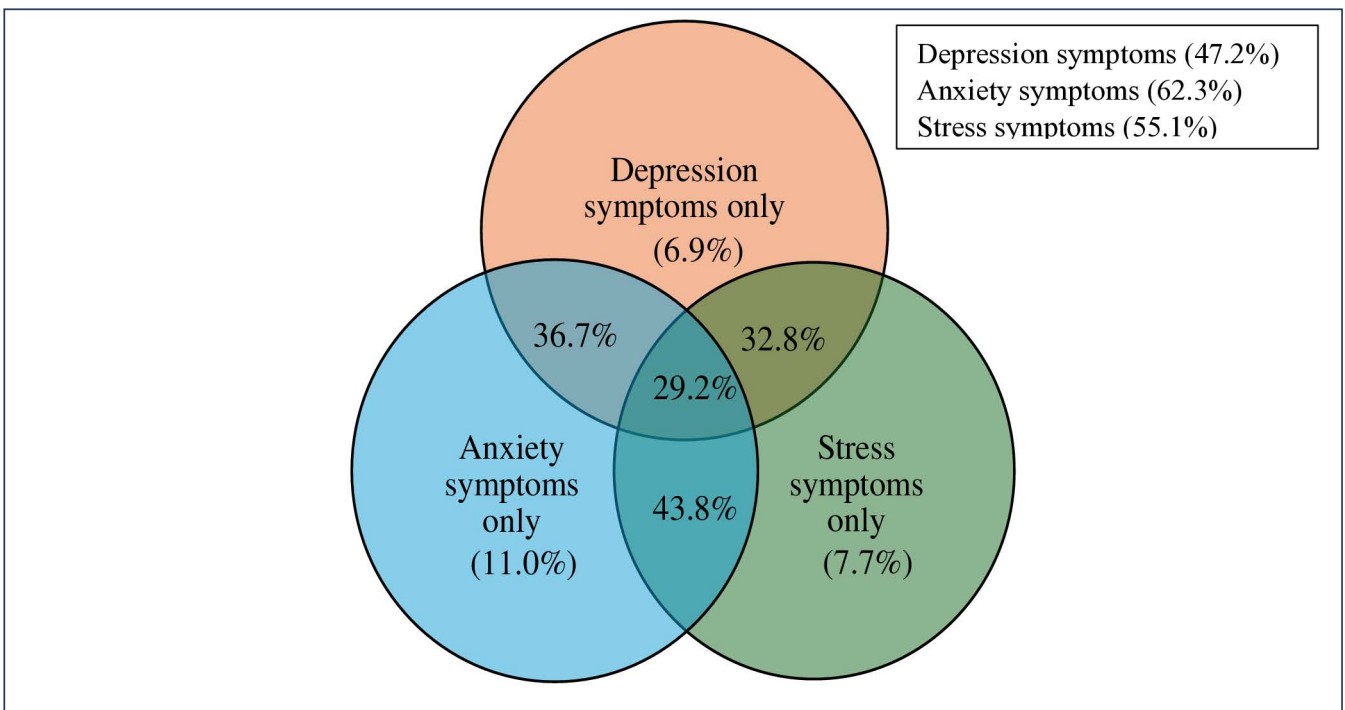

**Fig 2. Prevalence of depression, anxiety and stress symptoms (n = 390).**

for this association might be that longtime exposure to secondhand smoke can cause shortness of breath, and raise heart rate, potentially intensifying anxiety symptoms. Contrast to the findings, Thagunna and his colleagues showed a null association of ethnicity with stress, but with depression, among tribal Janajatias compared to Brahmin/Chhetri (non-tribal), among 16–40 years youth of nationally taken samples [30]. Possible reason for this contradiction is that depression among these underprivileged sub-populations are further mediated by stress (stressful events) including poverty and lack of social support, as revealed by Kohrt et al. [43].

In contrast to previous studies [22,23], smoking habits was found to be non-significant in this study. A possible reason for this negative association may be that few participants (16.9%) in this study were current smokers. Findings of previous studies showed that hypertension, and excess body fat were significantly associated with depression symptoms [8,40,44,45], but these are not significant in this study. The possible reasons for this negative association may be due to variation in study population, study settings, study variables, sample size, and sampling techniques. Another reason could be that CVDs risk factors are categorized using these variables. Since these variables are already included, they are found to be non-significant.

However, this study has some limitations as well that must be mentioned. First, quite less frequency of participants with zero CVDs risk factors were tapped, which may be cautiously interpreted when comparing. Secondly, binary logistic regression might have given less robust findings (to some extent). Thirdly, the study's reliance on self-reported data for identifying mental health status may have a potential for recall bias, to some extent. Finally, although the normality was checked, the large and limited (three) wards as clusters, and then conveniently selected samples from those wards might have limited the generalizability to some extent, and so, cautiously interpreted.

**Table 3. Regression analysis of depression symptoms.**

| #Variables | Depression (%) | cOR (95% CI) | p-value | aOR (95% CI) | p-value |
|---|---|---|---|---|---|
| **Sex** | | | | | |
| Male | 48 (37.2) | Ref | – | Ref. | – |
| Female | 136 (52.1) | 1.749 (1.137-2.691) | **0.006**** | 2.725 (1.575-4.715) | **<0.001***** |
| **Ethnicity** | | | | | |
| Brahmin/Chhetri | 149 (45.7) | Ref | – | Ref | – |
| Dalit, Janajati and Madhesi | 35 (54.7) | 1.546 (0.901-2.653) | 0.114 | 1.473 (0.814-2.667) | 0.200 |
| **Occupation** | | | | | |
| Unemployed | 45 (49.5) | Ref | – | Ref | – |
| Housekeeper, Farmer and Labor | 17 (35.4) | 0.951 (0.589-1.537) | 0.839 | 0.847 (0.470-1.526) | 0.581 |
| Private employee, Government job and Business | 17 (35.4) | 0.613 (0.300-1.253) | 0.180 | 0.631 (0.272-1.462) | 0.283 |
| **Consumption of Fruits and Vegetables** | | | | | |
| Low (≤1.3 servings/day) | 91 (51.1) | 1.338 (0.897-1.997) | 0.778 | 0.936 (0.589-1.485) | 0.778 |
| High (>1.3 servings/day) | 93 (43.9) | Ref | – | Ref. | – |
| **BP status** | | | | | |
| Normotensive | 114 (41.9) | Ref | – | Ref. | – |
| Hypertensive | 70 (59.3) | 2.021 (1.303-3.136) | **0.002**** | 1.456 (0.849-2.498) | 0.172 |
| **BMI status** | | | | | |
| Normal/underweight | 108 (40.8) | Ref | – | Ref. | – |
| Overweight/obese | 76 (60.8) | 2.255 (1.460-3.482) | **<0.001***** | 1.500 (0.743-3.028) | 0.258 |
| **Body fat status** | | | | | |
| Normal | 98 (42.2) | Ref | – | Ref. | – |
| Excess | 86 (54.4) | 1.633 (1.087-2.454) | **0.018*** | 0.899 (0.428-1.887) | 0.778 |
| **No. of CVDs risk factors** | | | | | |
| 0 risk factor | 9 (19.6) | Ref | – | Ref. | – |
| 1 risk factor | 43 (43.9) | 3.214 (1.401-7.375) | **0.006**** | 2.952 (1.224-7.121) | **0.016*** |
| 2 risk factors | 62 (49.6) | 4.046 (1.803-9.081) | **0.001**** | 3.904 (1.480-10.299) | **0.006**** |
| ≥3 risk factors | 70 (57.9) | 5.643 (2.503- 12.721) | **<0.001***** | 5.585 (1.357-22.991) | **0.017*** |
| **BMR** | | | | | |
| Lower | 123 (44.2) | 0.750 (0.495-1.135) | 0.168 | 1.309 (0.735-2.331) | 0.361 |
| Ideal | 61 (54.5) | Ref | – | Ref | – |
| **HGS** | | | | | |
| Lower | 61 (42.7) | 1.333 (0.881-2.019) | | 0.756 (0.460-1.242) | 0.269 |
| Ideal | 123 (49.8) | Ref | | | |

Significant at *p<0.05; **p<0.01; ***p<0.001; #DW statistic, 1.814; Tolerance, Min-Max, 0.258–0.927; VIF Min-Max, 1.078–3.883; Cox & Snell R Square, 0.112; Nagelkerke R Square, 0.149; -2 log likelihood, 493.216; Hosmer and Lemeshow test, 0.707.

## Conclusion

The study revealed nearly half of the participants have each of depression, anxiety, and stress symptoms. Inadequate intake of fruits and vegetables was observed among almost all, whereas overweight and obese and excess body fat were found among more than one-fourth. In comparison to zero risk factor, participants with two and three or more CVDs risk factors had a moderate-to-strong positive association with mental health status. Mental health screening is suggested

**Table 4. Regression analysis of anxiety symptoms.**

| #Variables | Anxiety symptoms (%) | cOR (95% CI) | p-value | aOR (95% CI) | p-value |
|---|---|---|---|---|---|
| **Sex** | | | | | |
| Male | 90 (69.8) | Ref. | – | Ref. | – |
| Female | 160 (61.3) | 1.629 (1.039-2.553) | **0.033*** | 1.338 (0.673-2.657) | 0.406 |
| **Ethnicity** | | | | | |
| Brahmin/Chhetri | 196 (60.1) | Ref | – | Ref | – |
| Dalit, Janajati and Madhesi | 47 (73.4) | 1.834 (1.009-3.333) | **0.044*** | 1.411 (0.736-2.703) | 0.300 |
| **Smoking status** | | | | | |
| Never or former smoker | 193 (59.6) | Ref | – | Ref | – |
| Current smoker | 50 (75.8) | 2.121 (1.158-3.885) | **0.013*** | 0.549 (0.176-1.714) | 0.302 |
| **Smokeless Tobacco** | | | | | |
| Never or former consumer | 180 (58.1) | Ref | | Ref | |
| Current consumer | 63 (78.8) | 2.676 (1.497-4.786) | **0.001**** | 1.817 (0.530-6.232) | 0.342 |
| **Secondhand smoke** | | | | | |
| No | 138 (55.0) | Ref | – | Ref | – |
| Yes | 105 (75.5) | 2.529 (1.597-4.005) | **<0.001***** | 2.065 (1.100-3.877) | **0.024*** |
| **BP status** | | | | | |
| Normotensive | 156 (57.4) | Ref | – | Ref | – |
| Hypertensive | 87 (73.7) | 2.087 (1.298-3.356) | **0.002**** | 1.086 (0.594-1.987) | 0.789 |
| **BMI status** | | | | | |
| Normal and underweight | 152 (57.4) | Ref | – | Ref | – |
| Overweight and obese | 91 (72.8) | 1.990 (1.252-3.162) | **0.004**** | 0.957 (0.436-2.102) | 0.913 |
| **Body fat status** | | | | | |
| Normal | 123 (53.0) | Ref | – | Ref | – |
| Excess | 120 (75.9) | 2.798 (1.790-4.375) | **<0.001***** | 1.499 (0.676-3.323) | 0.319 |
| **No. of CVDs risk factors** | | | | | |
| 0 risk factor | 10 (21.7) | Ref | – | Ref | – |
| 1 risk factor | 55 (56.1) | 4.605 (2.056-10.312) | **0.002**** | 4.703 (2.046-10.811) | **<0.001***** |
| 2 risk factors | 84 (67.2) | 7.376 (3.334-16.315) | **<0.001***** | 5.664 (2.190-14.646) | **<0.001***** |
| ≥3 risk factors | 94 (77.7) | 12.533 (5.515- 28.485) | **<0.001***** | 6.374 (1.314- 30.915) | **0.022*** |
| **BMR** | | | | | |
| Lower | 166 (59.7) | 1.484 (0.932-2.365) | 0.097 | 1.043 (0.567-1.918) | 0.893 |
| Ideal | 77 (68.8) | Ref | – | Ref | – |
| **HGS** | | | | | |
| Lower | 79 (55.2) | 0.625 (0.410-0.953) | **0.026*** | 0.644 (0.390-1.063) | 0.085 |
| Ideal | 164 (66.4) | Ref | – | Ref | – |

Significant at *p<0.05; **p<0.01; ***p<0.001; #DW statistic, 1.941; Tolerance, Min-Max, 0.243–0.935; VIF Min-Max, 1.070–4.112; Cox & Snell R Square, 0.143; Nagelkerke R Square, 0.194; -2 log likelihood, 456.736; Hosmer and Lemeshow test, 0.796.

among people with CVD risk factors, further gravitated towards depression, anxiety and stress; for female, secondhand smoker, and disadvantaged ethnic populations; respectively. Further, a stronger design incorporating appropriate number of participants with different cardiovascular risk statuses. We also further study for accuracy and yielding before making a firm conclusion.

**Table 5. Regression analysis of stress symptoms.**

| #Variables | Stress symptoms (%) | cOR (95% CI) | p-value | aOR (95% CI) | p-value |
|---|---|---|---|---|---|
| **Sex** | | | | | |
| Male | 81 (62.8) | Ref. | – | Ref. | – |
| Female | 134 (51.3) | 1.599 (1.039-2.463) | **0.032*** | 1.933 (0.444-1.961) | 0.855 |
| **Ethnicity** | | | | | |
| Brahmin/Chhetri | 172 (52.8) | Ref | – | Ref | – |
| Dalit, Janajati and Madhesi | 43 (67.2) | 1.994 (1.126-3.532) | **0.034*** | 2.085 (1.091-3.985) | **0.026*** |
| **Education** | | | | | |
| Literate | 179 (56.8) | Ref | – | Ref | – |
| Illiterate | 36 (48.0) | 1.547 (1.012-2.363) | 0168 | 0.623 (0.278-1.396) | 0.250 |
| **Occupation** | | | | | |
| Unemployed | 46 (50.2) | Ref | **–** | Ref | **–** |
| Housekeeper, Farmer and Labor | 137 (54.6) | 1.090 (0.674-1.761) | 0.726 | 0.926 (0.423-2.025) | 0.847 |
| Private employee, Government job and Business | 32 (66.7) | 2.060 (0.987-4.299) | 0.054 | 2.390 (0.822-6.947) | 0.109 |
| **Smoking status** | | | | | |
| Never or former smoker | 174 (53.7) | Ref | – | Ref | – |
| Current smoker | 41 (62.1) | 0.655 (0.379-1.131) | 0.129 | 0.548 (0.198-1.518) | 0.247 |
| **Smokeless Tobacco** | | | | | |
| Never or former consumer | 164 (52.9) | Ref | | Ref | |
| Current consumer | 51 (63.7) | 2.676 (1.497-4.786) | **0.048*** | 0.807 (0.251-2.596) | 0.719 |
| **Secondhand smoke** | | | | | |
| No | 126 (50.2) | Ref | – | Ref | – |
| Yes | 89 (64.0) | 1.766 (1.154-2.703) | **0.009**** | 1.778 (0.931-3.396) | 0.081 |
| **BP status** | | | | | |
| Normotensive | 132 (48.5) | Ref | – | Ref | – |
| Hypertensive | 83 (70.3) | 2.515 (1.586-3.988) | **<0.001***** | 1.327 (0.728-2.418) | 0.356 |
| **BMI status** | | | | | |
| Normal and underweight | 120 (45.3) | Ref | – | Ref | – |
| Overweight and obese | 95 (76.0) | 3.826 (2.376-6.162) | **<0.001***** | 1.430 (0.664-3.080) | 0.361 |
| **Body fat status** | | | | | |
| Normal | 96 (41.4) | Ref | – | Ref | – |
| Excess | 119 (75.3) | 4.323 (2.767-6.753) | **<0.001***** | 2.022 (0.909-4.499) | 0.084 |
| **Presence of CVDs risk factors** | | | | | |
| 0 risk factor | 6 (13.0) | Ref | – | Ref | – |
| 1 risk factor | 41 (41.8) | 4.795 (1.860-12.366) | **0.001**** | 6.119 (2.202-17.008) | **0.001**** |
| 2 risk factors | 76 (60.8) | 10.340 (4.079-26.212) | **<0.001***** | 7.800 (1.463- 41.582) | **0.016*** |
| ≥3 risk factors | 92 (76.0) | 21.149 (8.145- 54.916) | **<0.001***** | 8.484 (2.769-25.997) | **<0.001***** |
| **BMR** | | | | | |
| Lower | 133 (47.8) | 0.336 (0.208-0.524) | **<0.001***** | 0.736 (0.392-1.381) | 0.340 |
| Ideal | 82 (73.2) | Ref | – | Ref | – |
| **HGS** | | | | | |
| Lower | 63 (44.1) | 0.492 (0.324-0.748) | **0.001**** | 0.794 (0.468-1.347) | 0.392 |
| Ideal | 152 (61.5) | Ref | – | Ref | – |

Significant at *p<0.05; **p<0.01; ***p<0.001; #DW statistic, 1.751; Tolerance, Min-Max, 0.239–0.914; VIF Min-Max, 1.094–4.186; Cox & Snell R Square, 0.216; Nagelkerke R Square, 0.289; -2 log likelihood, 441.491; Hosmer and Lemeshow test, 0.684.

**Table 6. Predictors and their β-values (S.E.) of mental health status.**

| Covariates | Sex (female) | Ethnicity (dalit/ janajati/madheshi) | Secondhand smoking | CVDs risk factors | | |
| --- | --- | --- | --- | --- | --- | --- |
| | | | | 1 risk factor | 2 risk factors | ≥3 risk factors |
| Depression symptoms | 1.002 (0.002) ** | – | – | 1.082 (0.449) * | 1.362 (0.495) * | 1.720 (0.722) * |
| Anxiety symptoms | – | – | 0.725 (0.321) * | 1.548 (0.425) ** | 1.734 (0.485) ** | 1.852 (0.806) * |
| Stress symptoms | – | 0.735 (0.331) * | – | 1.811 (0.522) * | 2.054 (0.854) * | 2.138 (0.571) ** |

* P-value <0.05; ** p-value <0.01; Df for all covariates is 1.

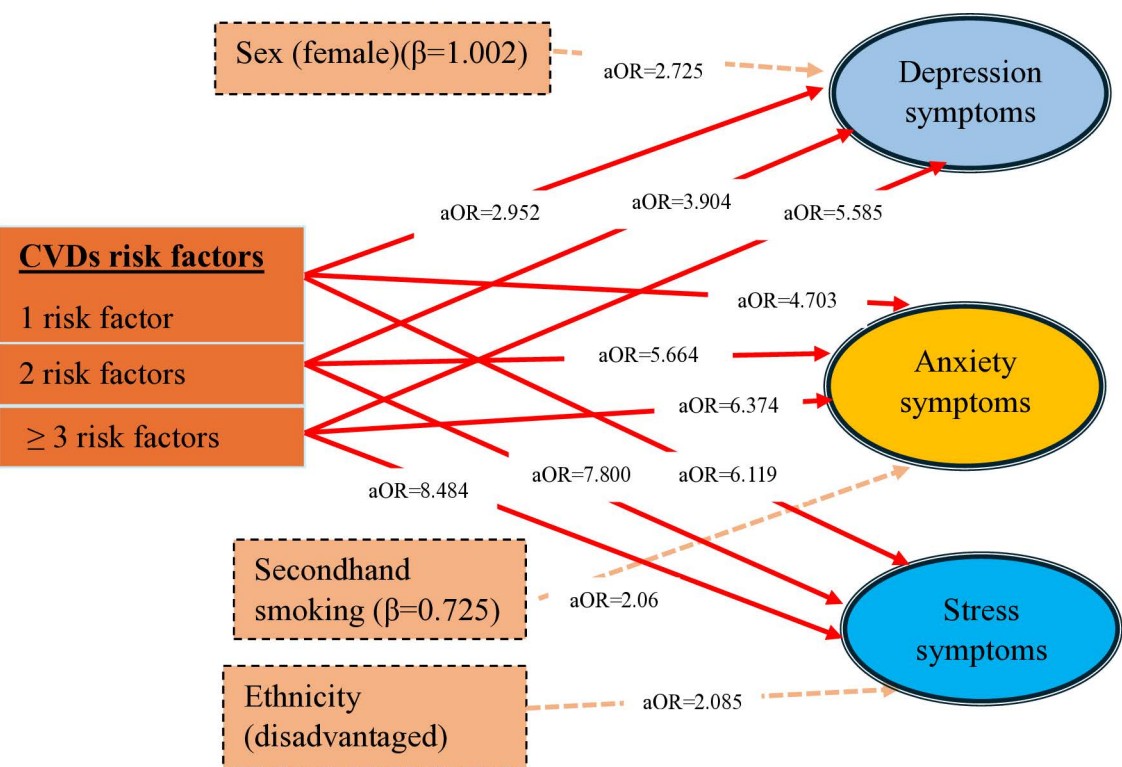

**Fig 3. Predictors of depression, anxiety, and stress symptoms.**

## Supporting information

**S1 Data. Set.**
(CSV)

**S1 File. S2 Table 1–3. Cross-tabulation of depression anxiety stress.**
(DOCX)

## Acknowledgments

We would like to acknowledge all participants who provided their valuable time for our study. We would like to express our gratitude to Mr. Basant Kumar Dhami and Mr. Ramesh Shrestha for their support in data analysis in our manuscript. Additionally, we are thankful to all those who provided us with direct and indirect support during our study.

## Author contributions

**Conceptualization:** Ramesh Ojha, Chiranjivi Adhikari, Hari Prasad Kaphle, Dikshya Adhikari.

**Data curation:** Ramesh Ojha.

**Formal analysis:** Ramesh Ojha, Chiranjivi Adhikari, Dikshya Adhikari.

**Investigation:** Ramesh Ojha, Chiranjivi Adhikari.

**Methodology:** Ramesh Ojha, Chiranjivi Adhikari, Hari Prasad Kaphle, Dikshya Adhikari.

**Project administration:** Ramesh Ojha, Chiranjivi Adhikari.

**Resources:** Ramesh Ojha, Chiranjivi Adhikari, Hari Prasad Kaphle, Dikshya Adhikari.

**Software:** Ramesh Ojha, Chiranjivi Adhikari, Dikshya Adhikari.

**Supervision:** Chiranjivi Adhikari.

**Validation:** Chiranjivi Adhikari.

**Visualization:** Ramesh Ojha, Chiranjivi Adhikari.

**Writing – original draft:** Ramesh Ojha, Chiranjivi Adhikari, Hari Prasad Kaphle, Dikshya Adhikari.

**Writing – review & editing:** Ramesh Ojha, Chiranjivi Adhikari.

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
