## [Decision Letter · Decision Letter 0]

31 Mar 2025

PONE-D-25-11173Intertwined risk factors of mental health and cardiovascular diseases: A Cross-sectional survey in Godawari Municipality of Far-western NepalPLOS ONE

Dear Dr. Adhikari,

Thank you for submitting your manuscript to PLOS ONE. After careful consideration, we feel that it has merit but does not fully meet PLOS ONE’s publication criteria as it currently stands. Therefore, we invite you to submit a revised version of the manuscript that addresses the points raised during the review process.

• The title is clear and reflects the study's focus; however, consider making it more specific to highlight the primary findings. Such as specifying CVD risk factors VS CVD outcomes and clarifying the “Intertwined” whether your objectives and analysis method enough to assess the Intertwined risk factors it          adequately. Revise your title accordingly. • Abstract: Typos “We data” – correct that. The abstract provides a concise summary of the study, but the results section could be expanded to better emphasize key findings and their implications. Please use word for indicating risk factors of CVD 1 CVD risk factor Vs one CVD risk factor for clarity. • Introduction: Consider incorporating additional recent literature to support the study's significance.• Further clarification on sample selection criteria and sample size calculation recommended. Since the objectives of the study was to assess… among “with and without CVD risk factors” population ( two groups) ; please clarify how these factors were incorporated in your sample size calculation and sample          selection? If not then how the validity of comparison of findings among these (among unequal groups) will be ensured. Please provide your justification.• Some sections could benefit from additional explanation, particularly in linking statistical outcomes to research questions. Consider addressing any potential biases in data interpretation.• In discussion, some areas could benefit from a more critical analysis of findings, particularly in comparison to similar studies.• The limitations section is acknowledged but could be elaborated further to discuss potential implications on study findings.• The conclusion summarizes key findings well but could more explicitly discuss policy or practice implications. Consider including future research directions based on study outcomes.  

We look forward to receiving your revised manuscript.

Kind regards,

Dr Buna Bhandari

Academic Editor

PLOS ONE

Additional Editor Comments:

• The title is clear and reflects the study's focus; however, consider making it more specific to highlight the primary findings. Such as specifying CVD risk factors VS CVD outcomes and clarifying the “Intertwined” whether your objectives and analysis method enough to assess the Intertwined risk factors it

adequately. Revise your title accordingly.

• Abstract: Typos “We data” – correct that. The abstract provides a concise summary of the study, but the results section could be expanded to better emphasize key findings and their implications. Please use word for indicating risk factors of CVD 1 CVD risk factor Vs one CVD risk factor for clarity.

• Introduction: Consider incorporating additional recent literature to support the study's significance.

• Further clarification on sample selection criteria and sample size calculation recommended. Since the objectives of the study was to assess… among “with and without CVD risk factors” population ( two groups) ; please clarify how these factors were incorporated in your sample size calculation and sample

selection? If not then how the validity of comparison of findings among these (among unequal groups) will be ensured. Please provide your justification.

• Some sections could benefit from additional explanation, particularly in linking statistical outcomes to research questions. Consider addressing any potential biases in data interpretation.

• In discussion, some areas could benefit from a more critical analysis of findings, particularly in comparison to similar studies.

• The limitations section is acknowledged but could be elaborated further to discuss potential implications on study findings.

• The conclusion summarizes key findings well but could more explicitly discuss policy or practice implications. Consider including future research directions based on study outcomes.

Reviewers' comments:

Reviewer's Responses to Questions

**Comments to the Author**

1. Is the manuscript technically sound, and do the data support the conclusions?

Reviewer #1: Yes

Reviewer #2: Yes

2. Has the statistical analysis been performed appropriately and rigorously? 

Reviewer #1: Yes

Reviewer #2: Yes

3. Have the authors made all data underlying the findings in their manuscript fully available?

Reviewer #1: Yes

Reviewer #2: Yes

4. Is the manuscript presented in an intelligible fashion and written in standard English?

Reviewer #1: Yes

Reviewer #2: Yes

5. Review Comments to the Author

Reviewer #1: Paper was very well written and easy to follow. All statistical analyses were performed and findings were available in the manuscript. This paper does not require heavy or major revisions. Very well done!

Reviewer #2: I thank the editors for inviting me to review the manuscript entitled Intertwined risk factors of mental health and cardiovascular diseases: A Cross-sectional survey in Godavari Municipality of Far-western Nepal by Chiranjivi Adhikari. very fascinating subject .The study is well designed and implemented .The aims and objectives were well defined The sample selection is well explained .Results were well depicted The discussion has nicely delt with each variable associated with CVD risk factors and mental health ,only one explanation is lacking showing, how the CVD risk factors

are directly related to increased prevalence of Depression anxiety etc .The explanation, why the female sex has more mental health problem than male sex is well explained but similar explanation regarding the increased prevalence of mental health problems with regard to CVD risk factors is lacking

6. PLOS authors have the option to publish the peer review history of their article (what does this mean? ). If published, this will include your full peer review and any attached files.

**Do you want your identity to be public for this peer review?** For information about this choice, including consent withdrawal, please see our Privacy Policy .

Reviewer #1: No

Reviewer #2: No

---

## [Author Response · Author response to Decision Letter 1]

17 Apr 2025

Comments and Responses from Editor and Reviewers, in point-by-point basis

Editor’s Comments

Comment 1:

The title is clear and reflects the study's focus; however, consider making it more specific to highlight the primary findings. Such as specifying CVD risk factors VS CVD outcomes and clarifying the “Intertwined” whether your objectives and analysis method enough to assess the Intertwined risk factors it adequately. Revise your title accordingly.

Response:

Thank you for raising a doubt that guided us for further clarification! ‘Intertwined’ has been applied to mean ‘mutually involved’ as from the literature most of them are established as “traditional risk factors”. Since the literature shows that they are still to be “established” in case of mental disorders, mainly anxiety, depression, and stress, and additionally, evidence from the country like Nepal, is scarce and so even more important. In this line, we tried to explain with those factors to be risk, also in case of the mental disorders. As a result, cumulatively, with the established evidence in case of CVDs, and here from our study with regression modelling, also found as risk factors of mental disorders, and came to a conclusion of the term—intertwined, for both, CVDs and mental disorders. In addition, there are similar topics, addressing the mutual involvements, can be found elsewhere in the literature. We still appreciate your specific feedback in this regard, if not clarified.

Comment 2:

Abstract: Typos “We data” – correct that. The abstract provides a concise summary of the study, but the results section could be expanded to better emphasize key findings and their implications. Please use word for indicating risk factors of CVD 1 CVD risk factor Vs one CVD risk factor for clarity.

Response:

Thank you for this suggestion. We spelled out the numbers 1-10 as per the guideline, and changed accordingly throughout the manuscript. We added participants, age, and sex descriptions. In addition, we also mentioned the relationship of increasing number of risk factors with worsened statuses of mental health disorders. Further implications have been explored in discussion section.

Comment 3:

Introduction: Consider incorporating additional recent literature to support the study's significance.

Response: Thank you for your comments, we have added 10 additional citations for justifying the significance, and to add to discussion (now total references; 45) with recent literature findings that support our study’s significance. We have carefully reviewed this aspect and added with following changes for significance:

There is a bidirectional relationship between mental health and CVDs.[8] Mental health issues are more common in people with cardiovascular disease and its risk factors than in healthy people[9] and severe mental illness has been linked to a two-fold increase in deaths from CVDs.[10] A prospective cohort study from the UK demonstrated that the highest mental health scores were associated with a 1.56-fold increase in CVD risk.[11] A systematic review found that mental health conditions like depression and anxiety may cause fluctuations in blood pressure, potentially leading to cardiovascular complications.[12] Research on mental health is often given less emphasize, despite the fact that depression and anxiety are predicted to cost the global economy $1 trillion a year.[13]

Evidence from recent studies showed that mental health is increasingly recognized as a major factor in cardiovascular disease risk and integrating mental health assessments into cardiovascular care is essential for reducing the disease's impact. A mendelian randomization carried out with a very large samples strongly evidenced that smoking is a causal factor for depression and schizophrenia.[14] Similarly, multi-cross-sectional findings from a cohort study of patients and their siblings also evidenced that smoking is associated with both positive and negative psychosis, and depression.[15] There are also strong findings from evidence syntheses. A metaanalysis showed 70% more odds of having severe mental illness (SMI), and it increases to 87% in case of schizophreniform illness, among the diabetics, and also, metabolic syndrome (MetS) is found in higher proportion among the people with SMI.[16] Inversely, another review showed that those with SMIs, 70% of their diabetic statuses remain unscreened.[17] Another recent cohort study The significant relationship of these two giant co-morbidities, and their risk factors, is complex, intertwined, multi-faceted, and magnitudinous, and further implicates healthcare delivery and outcomes, as in metaanalysis carried out by Ayerbe et al. revealed that odds of quitting smoking is 36% less among the depressed patients. [18]

We think now the significance is justified.

Comment 4:

Further clarification on sample selection criteria and sample size calculation recommended. Since the objectives of the study was to assess… among “with and without CVD risk factors” population ( two groups) ; please clarify how these factors were incorporated in your sample size calculation and sample selection? If not then how the validity of comparison of findings among these (among unequal groups) will be ensured. Please provide your justification.

Response:

We are truly thankful for the scientific and methodological point. We actually had carried out the cross-sectional survey and as from our PPS, we obtained different statuses of CVD risks, and from the obtained sample, we analysed. Being second to you, we have revised and replaced the term, “cardiovascular risk status” in place of “with and without CVD risk factors” throughout the manuscript

Comment 5:

Some sections could benefit from additional explanation, particularly in linking statistical outcomes to research questions. Consider addressing any potential biases in data interpretation.

Response:

Thank you for your comments, we have added additional explanations on methodology and discussion section addressing potential biases in data interpretation. We have made the following improvements;

We applied binary logistic regression to model the associations between CVD risk factors and mental health outcomes, accounting for potential confounding factors and providing adjusted odds ratios to quantify these relationships.

Thirdly, the study's reliance on self-reported data for identifying mental health status, introduces the potential for recall bias.

Comment 6:

In discussion, some areas could benefit from a more critical analysis of findings, particularly in comparison to similar studies.

Response:

Thank you for your valuable feedback regarding the discussion section of our manuscript. We appreciate your suggestion to incorporate a more critical analysis of findings, particularly in comparison to similar studies. We have carefully reviewed this aspect and made improvements with additional 10 citations with recent ones (including the significance)

Comment 7:

The limitations section is acknowledged but could be elaborated further to discuss potential implications on study findings.

Response:

Thank you for suggesting the need for a more detailed discussion of the study's limitations and their potential impact on our findings. We have expanded the limitations section to address this, including the following points:

The study's reliance on self-reported data for identifying mental health status which may create the potential for recall bias. Participants may underreport unhealthy behaviors or overreport symptoms, potentially affecting the accuracy of prevalence estimates and associations.

Finally, although the normality was checked, the large strata as municipality ward, only three out of 12, and then conveniently selected samples from those wards may have impacted the normality to some extent, and so, cautiously interpreted.

Comment 8:

The conclusion summarizes key findings well but could more explicitly discuss policy or practice implications. Consider including future research directions based on study outcomes.

Response:

Thank you for your valuable feedback regarding the conclusion section of our manuscript. We have added the future research directions in conclusion section.

Reviewers’ Comments

Reviewer #1

Comment 9:

Paper was very well written and easy to follow. All statistical analyses were performed and findings were available in the manuscript. This paper does not require heavy or major revisions. Very well done!

Response:

Thank you for your positive feedback and positive appraisal of the quality of our manuscript. Thank you for your time and consideration.

Reviewer #2

Comment 10:

I thank the editors for inviting me to review the manuscript entitled Intertwined risk factors of mental health and cardiovascular diseases: A Cross-sectional survey in Godavari Municipality of Far-western Nepal, very fascinating subject.

The study is well designed and implemented. The aims and objectives were well defined The sample selection is well explained. Results were well depicted. The discussion has nicely delt with each variable associated with CVD risk factors and mental health.

Response:

Thank you for your positive comment, and recognition of the quality of our manuscript.

Comment 11:

Only one explanation is lacking showing, how the CVD risk factors are directly related to increased prevalence of Depression anxiety etc. The explanation, why the female sex has more mental health problem than male sex is well explained but similar explanation regarding the increased prevalence of mental health problems with regard to CVD risk factors is lacking.

Response:

Thank you for your appreciation regarding why females have higher prevalences, especially, depression, in our case. We also appreciate your explorative inquisitiveness regarding why high (or low) prevalences of the three mental disorders associated with CVD risk factors. Our objective was to assess the prevalence of anxiety, depression, and stress; and further find out the risk factors of them, from the list of CVD risk factors, so to establish ‘intertwining’. However, as concerned to your query, further comparative design with geography/locations of high and low prevalences of the three mental disorders could help to point out the risk factors more robustly, but may only limiting to political geographical divisions or organization, which is out of the scopes of this study.

Nonetheless, we have tried to assess the same with cross-sectional design, explaining the CVD risk factors, such as obesity, excess body fat, and hypertension, the behavioral risk factors such as inadequate intake of fruits and vegetables, along with smoking and tobacco use, for anxiety, depression and stress. Pathways how these factors can impact mood, thinking, and emotional balance, leading to mental health issues, may be more complex, which are beyond the scope of the study.

With regards

Chiranjivi, Corr. Author

---

## [Editor Report · Decision Letter 1]

23 Apr 2025

Intertwined risk factors of mental health and cardiovascular diseases: A Cross-sectional survey in Godawari Municipality of Far-western Nepal

PONE-D-25-11173R1

Dear Dr. Adhikari,

We’re pleased to inform you that your manuscript has been judged scientifically suitable for publication and will be formally accepted for publication once it meets all outstanding technical requirements.

Kind regards,

Dr Buna Bhandari

Academic Editor

PLOS ONE

Additional Editor Comments (optional):

Please correct some grammatical and sentence errors during final proofreading of the article. Such as the second-to-last sentence in the conclusion, which is currently incomplete. "Further, a stronger design

incorporating appropriate number of participants with different cardiovascular risk status"
---

## [Editor Report · Acceptance letter]

PONE-D-25-11173R1

PLOS ONE

Dear Dr. Adhikari,

I'm pleased to inform you that your manuscript has been deemed suitable for publication in PLOS ONE. Congratulations! Your manuscript is now being handed over to our production team.

Kind regards,

on behalf of

Dr. Buna Bhandari

Academic Editor

PLOS ONE